# Auricular Acupuncture for Preoperative Anxiety—Protocol of Systematic Review and Meta-Analysis of Randomized Controlled Trials

**DOI:** 10.3390/medicines7120073

**Published:** 2020-11-26

**Authors:** Joanna Dietzel, Mike Cummings, Kevin Hua, Klaus Hahnenkamp, Benno Brinkhaus, Taras I. Usichenko

**Affiliations:** 1Institute of Social Medicine, Epidemiology and Health Economics, Charité University Medicine, 10117 Berlin, Germany; Joanna.dietzel@charite.de (J.D.); Kevin.Hua@charite.de (K.H.); benno.brinkhaus@charite.de (B.B.); 2British Medical Acupuncture Society, London WC1N 3HR, UK; Mike.Cummings@btinternet.com; 3Department of Anesthesiology, University Medicine of Greifswald, 17475 Greifswald, Germany; klaus.hahnenkamp@med.uni-greifswald.de; 4Department of Anesthesia, McMaster University, Hamilton, ON L8S 4K1, Canada

**Keywords:** auricular acupuncture, preoperative anxiety, protocol, randomized controlled trials, systematic review, meta-analysis

## Abstract

*Background:* Preoperative anxiety causes profound psychological and physiological reactions that may lead to a worse postoperative recovery, higher intensity of acute and persistent postsurgical pain and impaired quality of life in the postoperative period. Previous randomized controlled trials (RCTs) suggest that auricular acupuncture (AA) is safe and effective in the treatment of preoperative anxiety; however there is a lack of systematic evidence on this topic. Therefore, this protocol was developed following the PRISMA guidelines to adequately evaluate the existing literature regarding the value of AA for the reduction in anxiety in patients in a preoperative setting, compared to other forms of acupuncture, pharmacological, or no control interventions and measured with questionnaires regarding anxiety and fear. *Methods:* The following databases will be searched: MEDLINE (PubMed), EMBASE, Cochrane Central Register of Controlled Trials (CENTRAL), ISI Web of Science, and Scopus Database. RCTs will be included if an abstract is available in English. Data collection and analysis will be conducted by two reviewers independently. Quality and risk assessment of included studies will be done using the Cochrane 5.1.0 handbook criteria and meta-analysis of effectiveness and symptom scores will be conducted using the statistical software RevMan V.5.3. *Conclusions:* This systematic review will evaluate the efficacy and safety of AA for preoperative anxiety. Since all data used in this systematic review and meta-analysis have been published, this review does not require ethical approval. The results may be published in a peer-reviewed journal or be presented in relevant conferences. Registration number: PROSPERO ID CRD42020.

## 1. Introduction

Preoperative anxiety is the most frequent burden affecting patients before various surgical procedures [1]. Facing the prospect of surgery and hospitalization, patients experience fear, anxiety, uncertainty, loss of control and decreased self-esteem [2]. More than 90% of adult patients scheduled for elective surgery developed preoperative anxiety, and 40.5% reported severe anxiety [3]. Although the preoperative anxiety is a kind of situational anxiety, which terminates itself when the underlying condition (surgery) is over, it causes a profound psychological and physiological response via the release of stress-hormones, that may be associated with a worse postoperative recovery, higher intensity of acute and persistent postsurgical pain and greater anesthetic requirement as well as impaired quality of life in the postoperative period [4,5,6,7,8].

A large variety of approaches is used to treat preoperative anxiety including both psychological and pharmacological interventions; however, none of them seem to be ideal in providing effective, safe and low-cost therapy [9,10,11].

Auricular stimulation (including acupuncture and comparable techniques such as electroacupuncture and acupressure) is a method of complementary medicine, based on stimulation of cranial nerves. It has already been used to treat situational anxiety in experimental and clinical conditions [12]. In several randomised controlled trials (RCTs) evaluating a treatment of preoperative anxiety, auricular stimulation was superior to an array of control procedures, including placebo and sham interventions, and equally effective like premedication with benzodiazepines in patients scheduled for surgery under general anesthesia [13,14,15,16,17]. Moreover, this method was associated with fewer side effects compared with benzodiazepines, as well as a diminished physical stress reaction such as a reduced response of the autonomic nervous system [12,16,17].

The potential mechanism of auricular stimulation is attributed to the neuroanatomical conditions of external auricle. It is presumed that auricular stimulation exerts its anxiolytic effects via the involvement of cranial nerves [18], which leads to the modulation of the brain areas involved in the stress response, including the limbic system, locus coeruleus and hypothalamus [19,20,21].

Although the majority of RCTs on auricular stimulation for preoperative anxiety were in favor of this technique, these clinical investigations demonstrated a heterogeneity in regard to surgical procedures, control conditions and effect size, thus making it difficult to draw any definitive recommendations. It seems possible that auricular stimulation might serve as an effective replacement for insufficient conventional pharmacological premedication [9,22], thus more accurate estimation of the efficacy and safety of this complementary medicine intervention is needed. Therefore, this planned systematic review, including a meta-analysis of RCTs, will be performed to evaluate the effect size of auricular stimulation on preoperative anxiety applied alone or in addition to standard care in comparison with various control conditions. Data on the efficacy and safety of treatment will be calculated and summarized. The review will also try to identify the factors that may influence the effects of this intervention.

## 2. Materials and Methods

### 2.1. Eligibility Criteria for Including Studies in the Review

#### 2.1.1. Types of Studies

Only randomized controlled trials (RCTs) in European languages will be included. Results from quasi RCT will be discussed if little evidence is available, but they will not be part of the analysis. The funding source will be registered. Case reports, case-series, non-randomized case-control studies and retrospective data will not be included in the analysis.

#### 2.1.2. Types of Participants

No restrictions on study populations will be made, as long as they are described as patients undergoing surgical procedures, including all medical interventions requiring intra-procedural sedation or analgesia. There will be no restrictions regarding the age, gender or ethnicity of participants.

#### 2.1.3. Types of Interventions/Comparators

This review will include all studies, comparing auricular stimulation or related interventions (auricular acupuncture, auricular acupressure, auricular electroacupuncture, etc.) alone or in addition to routine care with a variety of control conditions, such as: sham acupuncture, sham acupressure, placebo, routine care, various cognitive-behavioral therapies (CBTs) such as relaxation techniques, music therapy, hypnosis, etc.

#### 2.1.4. Types of Outcome Measures

The primary outcome of this review will be the intensity of preoperative anxiety, measured using patient-reported psychophysical anxiety scales, such as the State Trait Anxiety Inventory (STAI), Anxiety Visual Analogue Scale-100 (VAS-100), the Amsterdam Preoperative Anxiety and Information Scale (APAIS), Self-Rating Anxiety Scale (SAS), extensively described elsewhere [23]. Since anxiety leads to the release of stress-hormones (i.e., vasoactive) the impact on physiological parameters will be collected and evaluated, being a surrogate parameter for pain and anxiety. Secondary outcomes will therefore include physiological parameters describing the response of the autonomic nervous system (e.g., heart rate, blood pressure, respiratory rate, sweating reaction); the preoperative requirement of anxiolytic medication; the intraoperative requirement of anesthetic and analgesic medication; the intensity of postoperative pain; the postoperative requirement for analgesic medication, the quality of blinding and patient-satisfaction with the treatment of preoperative anxiety.

#### 2.1.5. Safety of Intervention

Adverse events and serious adverse events reporting will be analyzed, including events such as pain, inflammation and infection at the sites of auricular stimulation, and vasovagal reactions during the auricular interventions.

### 2.2. Search Methods for Identification of Studies

The search will be done across the following electronic databases and registries, from their inception until June 2020: MEDLINE (PubMed), EMBASE, Cochrane Central Register of Controlled Trials (CENTRAL), ISI Web of Science, Scopus Database. The search terms will include: auricular, acupuncture, acupressure, preoperative anxiety, randomized clinical trials (Table 1).

### 2.3. Data Extraction and Management

#### 2.3.1. Study Identification

Two researchers will screen the titles and abstracts of articles found in the search, and discard trials that are not eligible. They will independently assess whether the trials meet the inclusion criteria, with disagreements to be resolved by discussion with a third researcher following objective criteria. When articles contain insufficient information to make a decision about eligibility, one of the researchers will attempt to contact the authors of the original reports to obtain further details via email. The details of data search and management are given as Figure 1. A new database for each of the two researchers will be set up to organize the data of the literature search.

#### 2.3.2. Data Extraction

Following the selection for inclusion, two researchers will independently extract data according to the standardized form designed by the review group (Table 2). A third researcher will check for accuracy and enter data into Review Manager software (RevMan 5.3. 2011).

#### 2.3.3. Assessment of Risk of Bias in Included Studies

Two researchers will assess all included trials for risk of bias, blind to each other’s assessments. Random sequence generation, allocation concealment, blinding of participants and personnel, incomplete outcome data, selective reporting and other potential sources of bias will be evaluated regarding low, high and unclear risk of bias according to Cochrane Collaboration assessment tool. Any disagreements will be resolved by discussion or by involving a third researcher to adjudicate.

#### 2.3.4. Measures of Treatment Effects

Since all outcome measures of this review represent continuous data, they will be presented as mean differences with 95% confidence intervals (CI), or as standardized mean differences (SMD).

#### 2.3.5. Dealing with Missing Data

All outcomes will be analyzed on an intention-to-treat basis. Corresponding authors from the trials with incomplete or insufficient data will be contacted via email to complete the data. Trials with greater than 20% of the data missing will be excluded from the analysis.

#### 2.3.6. Assessment of Heterogeneity

Statistical heterogeneity will be assessed in each meta-analysis using the *T*^2^, *I*^2^ and Chi^2^ statistics calculated by RevMan software. Heterogeneity will be regarded as substantial if *T*^2^ is greater than zero and either *I*^2^ is greater than 50% or there is a low P value (less than 0.10) in the Chi^2^ test for heterogeneity.

#### 2.3.7. Assessment of Reporting Biases

If the meta-analysis includes more than 10 investigations, reporting biases will be studied using a funnel plot with asymmetry testing.

### 2.4. Data Synthesis

Statistical analysis will be carried out using the RevMan software. Fixed-effect meta-analysis for combining data including primary outcome (anxiety scales) will be performed to estimate the treatment effect using SMD and 95% CI. In case of substantial clinical or statistical heterogeneity, a random-effects (RE) meta-analysis will be done to yield an overall summary. If RE analyses are necessary, their results will be presented as the average treatment effect with its 95% confidence interval, and the estimates of *T*^2^ and *I*^2^.

#### 2.4.1. Subgroup Analysis and Investigation of Heterogeneity

To assess potential heterogeneity, subgroup analyses will be performed including the following comparisons: adult versus pediatric patients; female versus male patients; emergency surgery versus elective surgery; inpatient versus outpatient surgery, as well as auricular acupuncture or auricular acupressure versus auricular acupuncture or acupressure plus other treatments (if applicable). The differences between subgroups will be assessed by interaction tests for fixed-effect inverse variance meta-analyses. For fixed-effect meta-analyses and RE using methods other than inverse variance, the comparison of subgroups’ confidence intervals will be used: non-overlapping confidence intervals indicate a statistically significant difference in the treatment effect between the subgroups.

#### 2.4.2. Sensitivity Analysis

Where subgroup analysis fails to explain the heterogeneity, data analysis using the RE model will be used. A priori sensitivity analyses on results will be done to look at the possible contribution of differences in methodological quality, comparing trials with a low risk of bias to all trials.

#### 2.4.3. Quality of Outcome Evidence

The quality of outcome evidence will be summarized using the Grading of Recommendations Assessment, Development and Evaluation (GRADE) approach. Each grade of evidence will be rated as: high, moderate, low or very low.

## 3. Discussion

Although almost 50% of adult patients scheduled for elective surgery suffer from preoperative anxiety [3], there is no ideal method to treat this kind of situational anxiety in clinical conditions so far. Pharmacological premedication is convenient in preoperative setting, however it seems to be less effective than previously suggested, if compared with placebo in trials with rigorous designs [9,22]. Psychological (cognitive-behavioral) approaches seem to be effective and lack dangerous side effects, however they are too time-consuming in their execution and thus are seldom used in routine clinical practice [24].

An array of data suggest that auricular stimulation might become such an effective, safe and easy-to-perform treatment for preoperative anxiety in adults scheduled for elective surgery and painful procedures with sedation [12,13,14,15,16,17]. Despite these promising results from clinical trials on the treatment of preoperative anxiety using auricular stimulation supported by neurophysiological explanation of its potential mechanisms [25], the systematic evaluation of the evidence for the treatment of anxiety using auricular acupuncture is not available.

This review and meta-analysis will fill this gap, analyzing the RCTs based on this protocol, which was designed according to the PRISMA statement (Appendix A). The review will calculate and summarise the data on the efficacy and safety of auricular stimulation in the treatment of preoperative anxiety in adult patients scheduled for elective surgery.

The results of this systematic review may be biased, since only the trials described in European languages will be considered, excluding the full format papers in native languages from the countries of the Far East, where auricular stimulation is widely used in traditional medicine [26]. Furthermore, the trials using transauricular vagal nerve stimulation (TaVNS) are considered to be beyond the scope of this review, despite the number of such trials growing rapidly in last two decades [27].

In summary, this systematic review will evaluate the existing evidence on the treatment of preoperative anxiety using auricular acupuncture and related procedures. The scheduled meta-analysis will estimate the effect of auricular stimulation on several perioperative parameters that are known to be influenced by preoperative anxiety. The results of this review will provide the basis for a better understanding of auricular acupuncture in the treatment of perioperative anxiety and will yield the evidence for the implementation of this method in clinical practice.

### Investigation Status

The data search is being performed for the present systematic review.

## Figures and Tables

**Figure 1 medicines-07-00073-f001:**
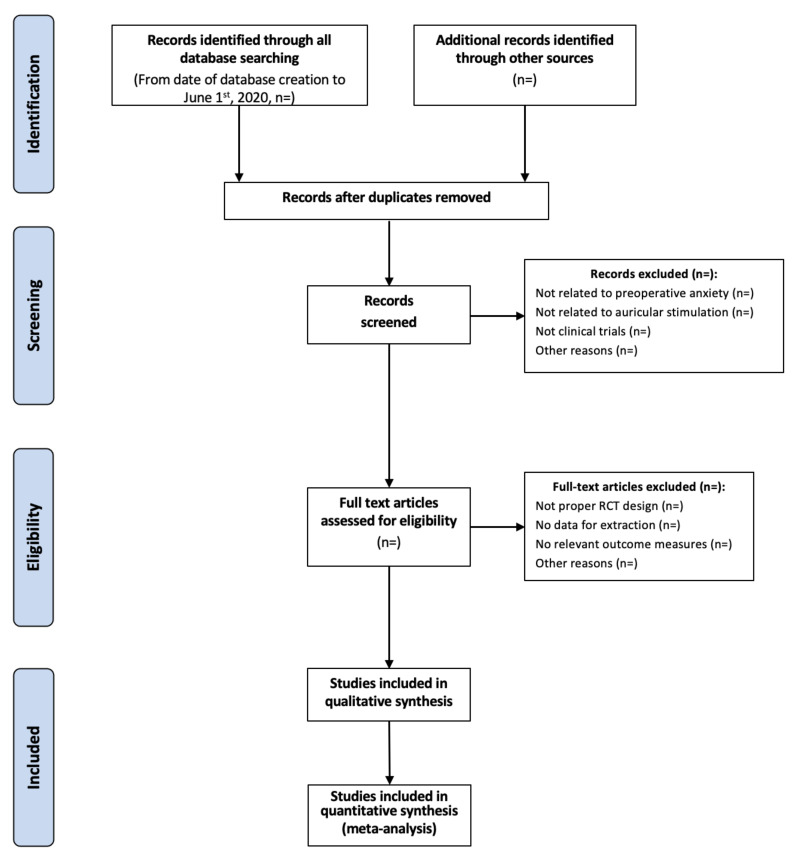
Flow diagram of details of data search and management.

**Table 1 medicines-07-00073-t001:** Search strategy used in MEDLINE database.

N	Search Item [Title/Abstract]
1	Randomized controlled trial
2	Controlled clinical trial
3	Randomized
4	Randomly
5	Trial
6	OR #1-6
7	Anxiety
8	Fear
9	Preoperative
11	Surgical
12	Intervention
14	Anesthesia
15	OR #7-14
16	Auricular acupuncture
17	Auricular
18	Ear
19	Acupressure
20	Electro-acupuncture
21	OR #16-20

This search strategy will be modified as required for other electronic databases.

**Table 2 medicines-07-00073-t002:** Content of data extraction.

N	Categories	Items Extracted
1	General information	Author, year of publication, title, journal (title, volume, pages), country, language of publication
2	Research method	Random allocation, allocation concealment, blinding, baseline level
3	Participants	Total sample size, number in experimental group, number in control group, gender, age, ethnicity, type of surgery, setting
4	Intervention	Type of intervention (auricular acupuncture, auricular acupressure, auricular electro-acupuncture, etc.), selection of auricular sites/auricular acupuncture points, selected for stimulation, type of device/needles, used for auricular stimulation, length of auricular stimulation, type of control condition
5	Outcome parameter	Levels of preoperative anxiety (taken using questionnaires and psychophysical scales), physiological parameters (heart rate, blood pressure, respiratory rate, sweating reaction, etc.), preoperative requirement of anxiolytic medication, intraoperative requirement of anaesthetic and analgesic medication, the intensity of postoperative pain, postoperative requirement of analgesic medication, patient satisfaction with the treatment of preoperative anxiety, safety and side effects of intervention and type of control condition

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
