# Peer review of "Auricular Acupuncture for Preoperative Anxiety—Protocol of Systematic Review and Meta-Analysis of Randomized Controlled Trials"

_medicines, 2020, doi:10.3390/medicines7120073_

Round 1

Reviewer 1 Report

The protocol for review is well organized and clearly lays out how the systematic review and meta-analysis of randomized controlled trials examining auricular acupuncture for preoperative anxiety will be done. The authors provide a good rationale for why systematic evidence on this topic is missing. Their methods for data extraction and management are well thought out and Figure 1 provides a visual summary of how the data search will be done. The authors present various outcome parameters they plan to look at through meta-analysis. As anxiety varies from person-to-person it is advantageous that the study will incorporate both objective measures of anxiety as well as physiological markers. Lastly, the authors have considered important parameters such as emergency surgery versus elective surgery and inpatient versus outpatient surgery to do subgroup analyses. This is favorable as it can help account for how the severity of the surgery could impact preoperative anxiety levels and treatment. Overall, this protocol is well written and takes into consideration a range of parameters impacting preoperative anxiety. The authors have clearly laid out how they plan to complete and troubleshoot this review should issues arise.   

Author Response

Thank you very much for your positive feedback.

Reviewer 2 Report

With respect to the manuscript “Auricular acupuncture for preoperative anxiety-protocol of systematic review and meta-analysis of randomized controlled trials”, I think that a systematic review in that subject would be interesting and important for professionals of the area.

However, I think that clarification about various aspects of the trial are needed:

  • Authors could define more precisely which are the objectives of the study: evaluate the existing literature, evaluate the effects of auricular acupuncture, calculate and summarize data on the efficacy and safety of treatment, identify the factors that influence the effects of the treatment.
  • The exclusion criteria should be explained more in detail.
  • Any doubt on the inclusion of a study should be resolved with a third researcher following objective criteria.
  • Data extraction should include possible side effects of the intervention and type of sham acupuncture.
  • For the subgroups analysis, I think it could be interesting comparing results from auricular acupuncture alone vs auricular acupuncture plus other treatments.

Author Response

Comment 1: Authors could define more precisely which are the objectives of the study: evaluate the existing literature, evaluate the effects of auricular acupuncture, calculate and summarize data on the efficacy and safety of treatment, identify the factors that influence the effects of the treatment.

Response 1: Since most of the above-mentioned objectives are listed at the end of the introduction (lines 59-63) we added a missing sentence in the introduction (lines 61-62) to address this comment: “Data on efficacy and safety of treatment will be calculated and summarized.”

Comment 2: The exclusion criteria should be explained more in detail.

Response 2: Thank you, we shave specified further, as recommended, in Materials and Methods Section, lines 69-70: “Case reports, case- series, non – randomized case- control studies and retrospective data will not be included in the analysis.

Comment 3: Any doubt on the inclusion of a study should be resolved with a third researcher following objective criteria.

Response 3: According to your recommendation we have changed the wording of the existing statement lines 110-111 to : ….with  a third researcher following objective criteria.

Comment 4: Data extraction should include possible side effects of the intervention and type of sham acupuncture.

Response 4: Since our control conditions are not only sham acupuncture, but as well sham acupressure we maintain the term control condition instead and added in table 2 Items under number 5: “…safety and side effects of intervention and type of control condition”

Comment 5: For the subgroups analysis, I think it could be interesting comparing results from auricular acupuncture alone vs auricular acupuncture plus other treatments.

Response 5: We extended the sentence (line 154-155) with: “….as well as auricular acupuncture or auricular acupressure versus auricular acupuncture or acupressure plus other treatments (if applicable).”

Reviewer 3 Report

Comments of authors
1. The impacts of anxiety during the surgery on human organs, hormones and the surgery success, please explain in more details.
2. Please, indicate to which degree the anxiety can be controlled using this protocol.
3. What are the used physiological parameters in registering the response of the nervous system while using this protocol?
4. Could you add all the actual sample size in numbers to figure 1.
5. Please emphasize if the two outcomes of the review were achieved.
6. Results, and conclusion are missing.
7. Authors could mention the outcome (advantages and disadvantages) secondary to applying this protocol.
8. Some abbreviations should be defined when appearing at first time.

Author Response

Comment 1.    The impacts of anxiety during the surgery on human organs, hormones and the surgery success, please explain in more details.

Response 1: According to your recommendation we extended the sentence in Introduction, line 34 as following : “…via the release of stress-hormones”. Further on, in Materials and Methods Section, lines 87-88 we have added a following sentence: “Since anxiety leads to the release of stress-hormones (i.e. vasoactive) the impact on physiological parameters will be collected and evaluated, being a surrogate parameter for pain and anxiety.”

Comment 2.    Please, indicate to which degree the anxiety can be controlled using this protocol.
Response 2: Since the actual manuscript contains the protocol for our planned systematic review and meta-analysis, no results or conclusions can be provided at this moment so far. These data will be published in the future paper.

Comment 3.    What are the used physiological parameters in registering the response of the nervous system while using this protocol?

Response 3: We focus on the surrogate parameters for anxiety such as heart rate, blood pressure, respiratory rate, sweating reaction (please see lines 83-94).

Comment 4.    Could you add all the actual sample size in numbers to figure 1.

Response 4: Since this is the protocol for our planned systematic review and meta-analysis, no sample size can be provided at this moment so far. These data will be published in the future result paper.

Comment 5.    Please emphasize if the two outcomes of the review were achieved.

Response 5: Since the actual manuscript contains the protocol for our planned systematic review and meta-analysis, no results or conclusions can be provided at this moment so far. These data will be published in the future result paper.

Comment 6.    Results, and conclusion are missing.

Response 6: Please, see Response 5.

Comment 7.    Authors could mention the outcome (advantages and disadvantages) secondary to applying this protocol.

Response 7: Unfortunately, at this moment we are not able to discuss the outcome, because our manuscript describes just a protocol of planned investigation. However, we will certainly do this in an actual meta-analysis paper.

Comment 8.    Some abbreviations should be defined when appearing at first time.

Response 8: We checked all abbreviations and they have been clarified at first mentioning.

Round 2

Reviewer 3 Report

Nothing